# A Proposed Saffron Soilless Cultivation System for a Quality Spice as Certified by Genetic Traceability

**DOI:** 10.3390/plants14010051

**Published:** 2024-12-27

**Authors:** Alessandro Mariani, Gianpiero Marconi, Nicoletta Ferradini, Marika Bocchini, Silvia Lorenzetti, Massimo Chiorri, Luigi Russi, Emidio Albertini

**Affiliations:** Department of Agricultural, Food and Environmental Sciences, University of Perugia, Borgo XX Giugno 74, 06121 Perugia, Italy; alessandro.mariani@dottorandi.unipg.it (A.M.); nicoletta.ferradini@unipg.it (N.F.); marika.bocchini@unipg.it (M.B.); silvia.lorenzetti@unipg.it (S.L.); massimo.chiorri@unipg.it (M.C.); emidio.albertini@unipg.it (E.A.)

**Keywords:** *Crocus sativus*, local varieties, SNPs, traceability, wooden bins

## Abstract

Saffron (*Crocus sativus* L.) is one of the most expensive spices in the world due to its strong market demand combined with its labor-intensive production process, which needs a lot of labor and has significant costs. New cultivation methods and traceability systems are required to improve and valorize local Italian saffron production. In this study, we conducted a three-year trial in Umbria (Central Italy), looking for a soilless cultivation method based on wooden bins posted at a suitable height from the ground to ease the sowing of corms and harvesting of flowers. Moreover, the spice traceability could be based on investigating the genetic variability of Italian saffron populations using SNP markers. The proposed novel cultivation method showed significantly higher stigma and corm production than the traditional one. At the same time, the genetic analysis revealed a total of 55 thousand SNPs, 53 of which were specific to the Italian saffron populations suitable to start a food traceability and spice certification.

## 1. Introduction

Saffron (*Crocus sativus* L.) is a triploid species probably originating in Attica, Greece, from *Crocus cartwrightianus* and selected by humans between 1600 and 350 B.C.E. [1]. Its spice, dried stigmas requiring extensive hard labor concentrated during the flowering period, is costly, making it the most expensive spice in the world [2,3]. Saffron spice has been used since ancient times in food, textile dye, cosmetics, or medicine [4], but in modern times, saffron is mainly used as a food coloring or flavoring. In recent decades, it has also been reconsidered for its medicinal properties as an antioxidant, anti-inflammatory, anti-apoptotic, hypotensive, hypolipidemic, antidepressant, anxiolytic, and neuroprotective [5,6,7].

The most important producer countries of saffron are Iran, India, Greece, and Afghanistan, with minor cultivation areas in Morocco, Spain, Italy, China, and Azerbaijan [8]. Italy has a long history of saffron cultivation, which began in the Roman era but declined during the barbarian invasions. In the Middle Ages, saffron cultivation was reintroduced in Italy by the Dominican Friar Santucci of Navelli (14th century A.D.), who planted in Abruzzo, Central Italy, corms brought from Spain and adapted the cultivation techniques to this area [9]. Nowadays, in Italy, saffron is mainly cultivated in Abruzzo (Piana di Navelli), Sardinia (Province of Medio Campidano), Tuscany (San Gimignano, the hills surrounding Florence and Maremma), and Umbria (Cascia and Città della Pieve), with three Protected Designations of Origin (L’Aquila, Sardinia, and San Gimignano) [10,11].

The area dedicated to saffron cultivation in Italy is limited and differs from that of the major producing countries. Overall, 70% of the Italian saffron production is grown in farms that dedicate less than 0.1 ha and only 6% in fields larger than 0.5 ha. Therefore, saffron production is also quantitatively limited: 44% of growers produce less than 100 g per year, 26% produce between 100 and 250 g, 17% produce between 250 and 500 g, 6% produce between 500 and 1000 g, and only 7% produce more than 1000 g per year [12]. The major constraint is finding labor during harvesting, so farming is based on available family human resources. Even though saffron cultivation is a side business for most Italian producers, Italian saffron is highly appreciated by the market for its high quality, with more than 80% of the samples analyzed classified in the first class according to ISO 3632 [13]. The main reason is due to the care for a limited amount produced and to the dehydration procedure that is conducted in electric ovens for few minutes at a relatively high temperature (80–90 °C), followed by a lower temperature (40 °C) able to maintain the quality [14,15]. For this reason, a traceability system for Italian saffron could protect the limited production and help prevent commercial fraud.

The biological cycle of saffron lasts about 200 days, from early autumn to May–June. Flowering begins in October and ends in November. In spring, corms start growing in size, reproduce, and enter a dormant state in summer. In Italy, saffron is mainly cultivated as an annual crop, unlike other countries where corms are usually left in the ground for three to eight years. Every year, corms are taken up in summer, selected for size, checked for defects such as rot and parasites [16,17], and replanted at the end of the warm season (August/September). This practice increases the production costs but allows for a better control of pests and increases corm size and weight compared to pluriannual crop systems. The yields in annual and pluriannual cropping systems were analyzed [17,18] and found to be higher in the biennial crop system, with a drastic reduction starting from the third year of cultivation.

The high costs of human resources and the difficulties in sowing and harvesting flowers (both non-mechanizable operations) render the area of saffron limited and confined to family management. In addition, open field cultivation is subject to pest and rodent attacks, further reducing the yield and economic return. For all the above reasons, we suggest exploring alternative cropping systems to ease the work and minimize yield losses. We recommend a system based on soilless cultivation in bins, which is less complex than the hydroponics and glasshouse techniques proposed by several authors [19,20,21,22,23,24,25,26].

The limited, high-quality Italian saffron production cannot compete with the world prices, as the rest of the world often produces spices at lower costs. For this reason, there is a need to organize farmers in associations and guarantee the origin and quality of the spice to consumers. For this purpose, molecular markers could be used as a valid tool to eventually find unique loci associated with a specific product and create a traceability system. The limited genetic variability of saffron populations, as reported by several authors [27,28,29,30,31] who investigated it by using different molecular markers, and due to its triploid nature and agamic propagation always by corms, could be better explored by SNPs markers, able to reveal differences even of a single nucleotide.

For all the above reasons, the main objective of this study was to assess the yield and quality of saffron spice, as well as corm yields, weight, and number by comparing the traditional cultivation method with plants grown in wooden bins posted at a suitable height from the ground to ease the planting of corms and harvesting of flowers. Additionally, the accessions from local farmers and other Italian accessions were characterized genetically and compared with samples from a World *Crocus* collection to look for private SNPs suitable for setting up a traceability system of Italian saffron.

## 2. Results

### 2.1. Field Trials

The combined ANOVA for all field traits is reported in Table 1. Significant differences were found among treatments for spice yield and length of the flowering period. The most productive combination was the cultivation in bins with organic soils (BO), whose mean was significantly higher than bins with farm soil (BF); this, in turn, was higher than the control field treatment. The spice yield in bins was higher despite a shorter flowering period length than the control treatment (Table 2).

No statistical differences among treatments were found for corm yield, which was measured as number and weight per unit area. Significant differences were instead found for the corm mean weight, with similar values in BO and control, both significantly higher than those in BF (*p* < 0.05). This was confirmed by the corm size (not reported because it was recorded in only three locations and for two years), whose mean diameters of 35.1 and 33.7 mm in BO and control were higher than 31.1 mm of BF, respectively (*p* < 0.05).

Differences among locations emerged only for the flowering period, with Alfonsi (32 days) significantly longer than Porta Sole, Ro_Lo, and Zafferano and Dintorni (22, 20, and 18 days). The shortest flowering period was found in Fontanelle, with an average of only 12.6 days.

The source of variation year was highly significant for all the above traits, indicating that the three seasons had an important impact on spice mean yield, with decreasing values from 2018 to 2020, respectively, on the number of flowers and flowering length, as well as in corm number, yield, and weight, with almost the same mean ranking trend as described for spice (Table 2). These differences were basically due to the discrepancies between seasons in terms of rainfall and temperatures and were in line with the yields recorded in the production area.

A significant treatments-by-location interaction was found only for the total number of flowers per unit area, and it was indicated that the response of the three cultivation treatments was different in different locations (Figure 1).

The interactions of year with all other sources of variation were highly significant (Table 1), indicating a different behavior of location and treatments in different years.

The saffron spice was of high quality in terms of moisture content, bitterness, aroma, and color strength. Significant differences in moisture content were found for treatments and locations (Table 3), although the means among treatments and locations were of little entity and all below 8% (from 6.58 to 7.96).

The differences among locations for bitterness showed the highest value in Fontanelle and Ro_Lo and the lowest in Mazzuoli and Alfonsi, with all the rest ranking at intermediate positions (*p* < 0.05). Considering that the range of values to classify the saffron spice in the top category (Type I) are moisture content below 10%, bitterness values higher than 70, aromatic strength between 20 and 50, and coloring strength higher than 200, the mean values reported in Table 4 are all higher than the standards, and it is possible to conclude that treatments and location did not have any impact on saffron quality.

### 2.2. Population Genetics

#### 2.2.1. SNPs Identification

The raw sequencing data from 65 samples were processed using the Stacks v.2.55 pipeline to demultiplex and clean the reads [32], resulting in a VCF file containing 386,009 SNP variants. Variants were filtered using two primary criteria: missing data and Minor Allele Frequency (MAF). Variants with more than 20% missing data across the samples were excluded. An MAF filter of 0.1% was applied to refine the dataset by retaining even rare mutations that may have arisen in specific populations and been maintained through asexual reproduction. After using these filters, the final dataset consisted of 54,711 variants across the 65 samples, with 7.3% missing data remaining. This process ensured 14.2% of polymorphisms, representing an informative set of SNP variants suitable for downstream analysis.

#### 2.2.2. Population Structure and Genetic Variability

The genetic structure of saffron populations was analyzed using Discriminant Analysis of Principal Components (DAPC) with the R package adegenet [33,34]. The analysis was based on 34 Principal Components (PCs), which retained 68.8% of the genetic variance, and two discriminant functions (DFs) (DF1 62.48%, DF2 2.57%) (Appendix A). Considering the DF1 coordinate, which explains the majority of variance, only the France and Argentina genotypes showed a relevant differentiation from the other accessions. The remaining saffron populations did not show distinct differences and were, therefore, considered a single group. Nevertheless, the DF2 coordinates provided slightly more insights into the relationship between genetic diversity and geographical origin, distinguishing between Western (Italy/Spain) and Eastern (Greece/Turkey/Iran/India) Mediterranean collections. In particular, the Italian and Spanish accessions formed an overlapped cluster, indicating low intra-group genetic diversity and a homogenous genetic background.

Furthermore, genotypes from Turkey and Iran exhibited some overlap, pointing out a shared genetic background likely influenced by their geographical proximity. Finally, as expected, looking at the DF2 coordinate, the Greek and Indian groups displayed a vicinity and a consistent genetic divergence from the others. The DAPC plot (Figure 2) examined the population structure more closely, highlighting subtler genetic relationships.

To further examine the genetic relationships, a membership probability plot was generated using the compoplot function from adegenet (Figure 3). This plot showed that individuals from the Argentinian and French populations were exclusively assigned to their respective genetic clusters, confirming the DAPC analysis and indicating a close genetic structure. In contrast, some individuals from all other countries displayed mixed probabilities, indicating varying levels of admixture or shared genetic traits. Individuals from the Italian population were predominantly assigned to a unique genetic cluster, indicating a well-defined genetic background with minimal evidence of admixture from other origins, mainly from Spain (in red) and, in one case, from Argentina (light orange).

The estimation of genetic differentiation between saffron populations was assessed using FST values based on different degrees of heterozygosity. The pairwise FST values ranged from 0.004 to 0.021 (Figure 4), revealing low levels of differentiation across the eight populations analyzed.

The highest average differentiation values were observed for Argentina (0.0154), India (0.0157), and France (0.0127). In contrast, the remaining populations showed very low average differentiation, ranging from a minimum of 0.0091 (Italy) to a maximum of 0.0101 (Spain).

#### 2.2.3. Private Alleles

The analysis of private alleles across saffron populations highlighted genetic differentiation. The Italian population exhibited the highest count of unique private alleles, with a total of 15,552, followed by Iran (5395 private alleles), France (4481), and Spain (3379). The Greek population exhibited 2841 unique alleles, followed by Argentina (1306). Turkey (1002) and India (1162) had the fewest number of private alleles. To identify the most frequent alleles for each population, we considered only those present at least three times (common to the three samples), thus obtaining a subset of informative private alleles suitable for potential genetic traceability in saffron. Applying this criterion, we narrowed down the private alleles to 53 in the Italian accessions, 33 in the Greek, 27 in the Iranian, 16 in the Turkish, 12 in the Indian, 6 in the French, 3 in the Spanish, and 2 in the Argentinian populations.

## 3. Discussion

### 3.1. Field Trials

In Italy, saffron is basically cultivated as an annual crop, with corms removed from the soil in June, cleaned and inspected for diseases, and re-planted in September. This system’s advantage is to rotate it with other crops and, hence, reduce losses due to disease outbreaks and weed competition. The disadvantages are increased costs due to soil preparation and the need for annual planting operations. However, it is widely recognized [2,35] that the main constraint for saffron cropping systems is the time needed to harvest the flowers daily and process them as soon as possible. In addition to this, corm planting and flower harvesting are laborious and not viewed as desirable by agricultural workers due to the hard-working conditions of the operators. All the above reasons have a negative impact on saffron production in Italy, as well as in some other Mediterranean countries [36]. To reverse this trend, several authors have suggested growing saffron in soilless systems (in peat and/or perlite substrates, as well as in hydroculture) in cold glasshouses or growth chambers [19,20,21,22,23,24,25,26]. In the present work, we investigated the possibility of growing saffron in wooden bins placed in an open field at a height suitable for the operators to facilitate the planting and harvesting operations. Over three years, the spice yield in BO was significantly higher than those in BF, and this was higher than the control (*p* < 0.01). The BO increase over the control was 62%, and that of BF was 35%. These differences seemed to be due to a higher number of flowers (138.5 vs. 126.2 m^−2^) rather than a longer flowering period (19.4 vs. 22.2 days, respectively, *p* < 0.05), suggesting an additional economic advantage in terms of saving costs of harvesting, flower processing, and pistil drying. In our context, spice yield and flower number were slightly lower than those reported by Maggio et al. (2006) [20], but still interesting considering our lower corm rate and less intensive growing conditions (glasshouses or growth chambers).

Corm size and weight were positively influenced by the control treatment’s deep soil condition, whose roots could explore and benefit from more available nutrients. In fact, corm size and weight were significantly lower in BF treatment. Interestingly, despite the limited bin depth of 30 cm, BO corm dimensions and weight were similar to the control and could have been advantaged not only by the available nutrients but also by the lower compactness of the organic-rich substrate. In our context, corm dimension and weight were higher than those reported by Maggio et al. (2006) [20], and this could be due to our lower corm rate and less intensive and competitive growing conditions.

Although the project did not include an in-depth analysis of the economic profitability of saffron cultivation in wooden bins, the analysis of the technical data concerning its cultivation suggests interesting economic potentialities for a number of reasons. Firstly, there would be a significant reduction in labor requirements compared to open-field cultivation, even if the data arise from experimental trials. Moreover, various operations (i.e., corm transplanting and flower harvesting, not otherwise mechanizable and concentrated in the early stage of the crop) are easier and hence faster, being performed in an upright rather than prone position (Figure 5). In fact, the concentration and timing of operations in the traditional cropping system during these two periods limit the area dedicated to saffron and, consequently, the quantity of spice that can be produced. It should also be considered that almost all producers are family-run, with no or few employees, and, therefore, the quantity produced is limited. The last advantage of cropping saffron in raised wooden bins is reduced losses caused by wild animals, mainly rodents, that normally feed on corms [37,38]. In our three years of experience, corms in bins were never damaged, while those in nearby open fields were. The disadvantage of the proposed system is the need to invest in bins and in their management; however, the reduction in labor requirements looks profitable, allowing for significant savings.

### 3.2. Population Genetics

The genetic analysis of saffron populations in this study revealed genetic clusters with no evidence of admixture in regions such as France and Argentina. In contrast, populations from Italy, Spain, Turkey, and India displayed varying levels of genetic mixing. These findings align with previous research, indicating that there is some genetic diversity within specific geographical regions despite saffron’s predominant vegetative propagation. Studies using molecular markers showed that saffron, although often considered a monomorphic species [39,40], can exhibit polymorphism and genetic variability. For instance, by analyzing Iranian saffron landraces using ISSR and SSR markers, researchers identified distinct genetic structures and moderate levels of variability, with notable admixture in some populations [27,28]. Similarly, studies on Indian populations using RAPD and ISSR markers detected genetic differentiation [29]. Siracusa et al. (2012) [31] used AFLP markers to distinguish European populations from those outside Europe. Consistent with findings reported by Busconi et al. (2021) [41], who analyzed genetic and epigenetic variability of saffron samples from India and Spain, demonstrating the presence of genetic variability within this species using SNPs, our results confirm that these markers can effectively discriminate the geographic origin of saffron.

The overall FST value of 0.006 indicates a very low genetic differentiation between populations, and this agrees with the triploid nature of the species whose reproduction is only via corm propagation. In addition, the pairwise FST values (Figure 4), ranging from 0.004 to 0.021, indicate low genetic variability, supporting the hypothesis of a shared genetic background among these saffron populations. These results contrast with another study [42], which investigated the genetic variability using SSR markers among several *Crocus* species, including *Crocus sativus* accessions. The authors reported higher pairwise FST values when comparing different *C. sativus* accessions, probably due to a limited number of polymorphic loci considered in their analyses. However, in our study, the relatively higher differentiation observed in populations such as those from India, France, and Argentina using SNP markers may reflect mutations raised independently in the accessions and/or to selection for local adaptation, leading to unique genetic profiles spread out through vegetative reproduction.

The use of DNA barcoding has been explored as a tool to identify the geographical origin of saffron spice and to detect potential adulterants [43,44,45]. Building on this approach, we examined private alleles to assess their effectiveness in distinguishing the origin of saffron. Our findings showed a high abundance of private alleles in the Italian population, likely due to the larger sample size of Italian accessions compared to other populations. Nevertheless, these private alleles may serve as valuable markers for the genetic traceability of Italian saffron. Since saffron spice consists of a bulk of dried stigmas from several plants, a combination of specific private alleles in the Italian population could serve as an effective strategy to detect the presence of Italian genotypes within the bulk. However, it is important to note that to fully validate these findings, additional accessions from other geographic origins should be included in future analyses.

## 4. Materials and Methods

### 4.1. Field Trials

#### 4.1.1. Cultivation Methods

The experiment consisted of assessing spice yield (g m^−2^), quality, and corm production (number and weight per m^−2^) from plants grown in open field plots, as in traditional cropping systems, compared with plants grown in wooden bins. In addition, since saffron is sensitive to waterlogging, bins were filled with farm soil (BF) or a loose loam rich in organic matter (BO). The bin treatments were compared with saffron corms grown in open field conditions (control). All corms were certified and provided by Mazzuoli farm to ensure a similar genetic material, selected for similar dimensions (28 ± 2 mm) and weight (20 ± 1 g).

The three treatments (BO = bins with organic soil, BF = bins with farm soil, and CL = control field plot) were replicated four times and arranged in a randomized complete block design. Bins were made of fir pre-treated with anti-rotting products. Their dimensions (200 × 40 × 30 cm in length, width, and depth) ensured an area of 0.8 square meters and a volume of 0.24 cubic meters of substrate. The bins were placed on 70 cm high trestles to facilitate sowing and harvesting operations (Figure 5).

The soil in the BO treatment was prepared with Irish fine peat (40%), silica sand (25%), volcanic pumice (25%), and organic matter (10%), all added with organic fertilizer (1 kg m^−3^). The soils of local farmers are typical of Central Italy, clay and loamy, and poorly drained, so in traditional cultivation, the fields are first prepared in hills, and corms are planted by hand in the ridge of the hills. The comparison of BO and BF was expected to provide information on the response of corms grown in bins with different substrates; the comparison of BF and control was expected to provide information on the innovative technique.

The study was carried out in eight small-scale farms specialized in saffron cultivation and located in seven areas scattered throughout the province of Perugia (Umbria, Central Italy; Table 5). The experiments were conducted for three consecutive years, from 2018 to 2020. Each season, the corms were planted in early September and removed from the field in early July of the following year.

Corms were planted at a density of 70 corms m^−2^, typical of the area. At the onset of flowering (October), flowers were harvested early in the morning every day, counted, and brought to the laboratory. The styles and stigma were removed and immediately dried in ovens at 80 °C for 3–5 min and at 60 °C for the time necessary to completely dehydrate the spice. The daily spice production was collected in glass jars, hermetically closed, and stored in the dark and at room temperature.

At the end of each harvesting season, the yield of each treatment was assessed, and, due to the limited amount of spice per treatment, a single, bulk sample of the four replicates was sent to a laboratory and analyzed for (1) moisture content, (2) aromatic strength, (3) bitterness, and (4) color strength. Saffron samples were quantified for picrocrocin, considering the molecular coefficient absorbance at 257 nm (bitterness); for safranal at 330 nm (aromatic strength); and for crocin at 440 nm (coloring strength). Analyses were performed in duplicates according to ISO 3632-2:2010. Moisture was also determined according to ISO 3632. Due to the limited amount of spice available per treatment, the statistical analyses were carried out using the data from each year as replication.

At the end of each vegetative season, corms were removed from the soil, cleaned, inspected for the presence of diseases, counted, and weighed. Each year, a new set of corms was used.

#### 4.1.2. Statistical Analysis

Each year’s data and location were analyzed first separately and later in a combined ANOVA. The combined model accounted for the main effects (treatment, location, and year), as well as their interactions. Year was considered a random factor, while location and treatments were considered fixed. The GLM used for the analysis was as follows:Y_ijk_ = μ + T_i_ + L_j_ + Y_k_ + (TL)_ij_ + (TY)_ik_ + (LY)_jk_ + (TLY)_ijk_ + e_ijk_
where

Y_ijk_ is the yield or quality parameter of spice or corms for the i-th treatment (T_i_), j-th company (L_j_), and k-th year (t_k_);μ is the grand mean;T_i_ is the effect of the i-th treatment;L_j_ is the effect of the j-th location;Y_k_ is the effect of the k-th year;(TL)_ij_ is the interaction effect between the i-th treatment and the j-th company;(TY)_ik_ is the interaction effect between the i-th treatment and the k-th year;(LY)_jk_ is the interaction effect between the j-th company and the k-th year;(TLY)_ijk_ is the interaction effect between the i-th treatment, the j-th company, and the k-th year;e_ijk_ is the random experimental error associated with each observation.

### 4.2. Population Genetics

#### 4.2.1. Plant Material

To genetically differentiate Italian accessions from those originating from other countries, we included additional plant material sourced from different locations in the world. Specifically, these materials were obtained from the World Saffron and *Crocus* Collection (WSCC) at the Germplasm Bank of Cuenca (Spain), from the experimental fields of the Department of Agricultural, Food, and Environmental Sciences of Perugia (DSA3); other corms were kindly provided by Dr. Busconi from the Faculty of Agricultural, Food, and Environmental Sciences at Università Cattolica del Sacro Cuore, Piacenza, and Dr. Yousefi from the Department of Plant Production, Faculty of Agriculture, University of Torbat Heydarieh, Torbat Heydarieh, Iran (Appendix A).

#### 4.2.2. Genomic Library Preparation, Sequencing, and Genetic Analysis

The leaves of saffron accessions were collected when they reached a length of 10 cm, placed in separate bags labeled with sample codes, and stored at −80 °C in the Laboratory of Agricultural Genetics and Biotechnologies at DSA3. DNA extraction was performed using the commercial kit GenElute™ Plant Genomic DNA Miniprep (Sigma Aldrich, St. Louis, MO, USA). The extraction procedure followed the standard protocol recommended by the manufacturer. The extracted DNA’s concentration and quality were verified through agarose gel electrophoresis (1.0%) and subsequent spectrophotometric reading using NanoDrop™ 2000 (Thermo Scientific, Waltham, MA, USA). Finally, the samples were quantified using a Qubit fluorometer (Invitrogen, Waltham, MA, USA) before NGS library preparation. Libraries were performed as described in the MCSeEd method [46]. Briefly, starting with 300 ng of genomic DNA, double digestion was carried out with a methylation-sensitive restriction enzyme (PstI) and a methylation-insensitive restriction enzyme (MseI). Pooled libraries were purified, size-selected (250–600 bp), quantified, normalized, and indexed for demultiplexing (Appendix A). Unique indices were added, followed by PCR enrichment and 150 bp paired-end Illumina sequenced (Novogene, Berkeley, CA, USA). Illumina raw reads (with the Q > 30) from libraries were demultiplexed and filtered (Appendix A) using the process_radtags utility included in the Stacks v2.0 [32] (scripts are available online at https://bitbucket.org/capemaster/mcseed/src/master/, accessed on 18 November 2024). Briefly, raw reads were processed using Rainbow 2.0.4 (https://sourceforge.net/projects/bio-rainbow, accessed on 18 November 2024) [47] and CDHit (https://github.com/weizhongli/cdhit, accessed on 18 November 2024) [48] to generate a pseudo-reference genome in the form of a multi-fasta file. These reads were then aligned to the pseudo-reference using *bwa mem* [49], and the mapping results were analyzed with Samtools to produce a count matrix. Variant calling was performed using the Stacks Suite v2.0 (gstacks program) [32], which involved creating a population file and a catalog of mutations. This process resulted in a VCF file containing allele frequencies and reference/alternative alleles. Variants were filtered based on missing data and Minor Allele Frequency (MAF), excluding those with more than 20% missing data and an MAF below 0.1%. The population’s genetic structure was analyzed using functions from the adegenet R package [50]. Specifically, the dapc function was applied for the Discriminant Analysis of Principal Components (DAPC). In contrast, the optima function was used to retain 34 Principal Components, as determined by the calculated best a-score (Appendix A), along with two discriminant functions [33,34]. The scatter, compoplot, and distgenepop functions were subsequently utilized to generate DAPC plots, calculate membership probabilities, and compute Fst pairwise values, respectively. For identification of private alleles, the “poppr” R package [51] was used employing the function private.allele.

## 5. Conclusions

Our study demonstrates how raised wooden bins offer a practical alternative to traditional saffron cultivation, increasing yields, reducing labor, and preventing rodent damage despite requiring initial investment. Genetic analysis confirmed low variability among saffron populations but identified unique traits in some regions, highlighting opportunities for genetic traceability. These findings suggest that improved methods and further research can enhance saffron production’s efficiency and sustainability.

## Figures and Tables

**Figure 1 plants-14-00051-f001:**
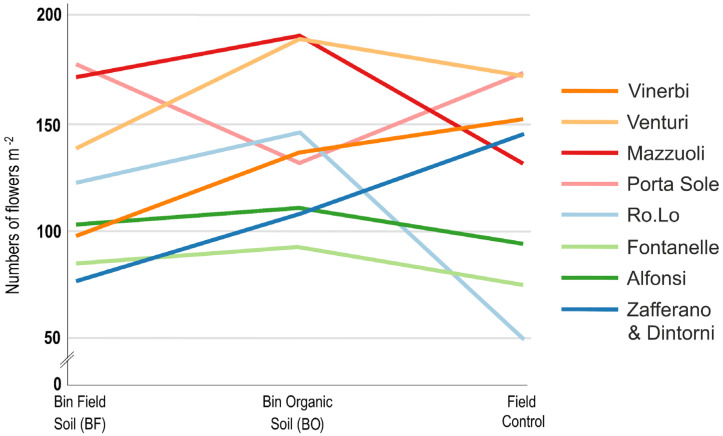
Treatment-by-location interaction for the number of flowers harvested per square meter from BF, BO, and field control.

**Figure 2 plants-14-00051-f002:**
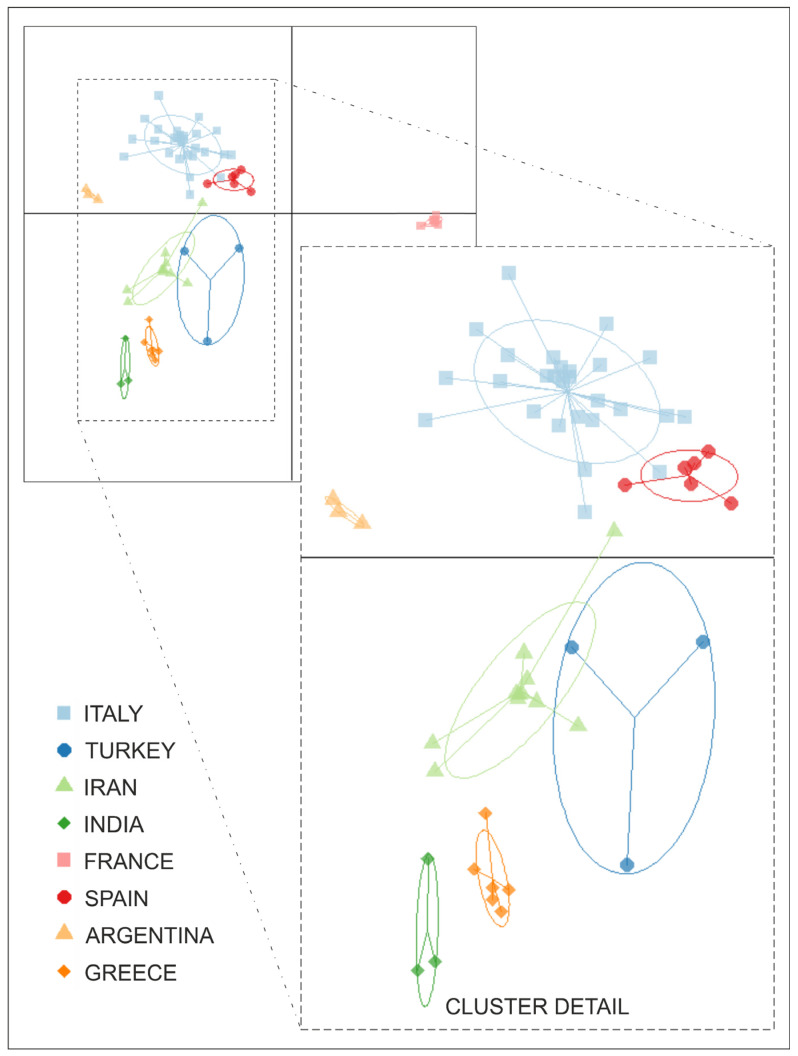
Scatter plot of Discriminant Analysis of Principal Component performed retaining 34 PCs and 2 DFs.

**Figure 3 plants-14-00051-f003:**
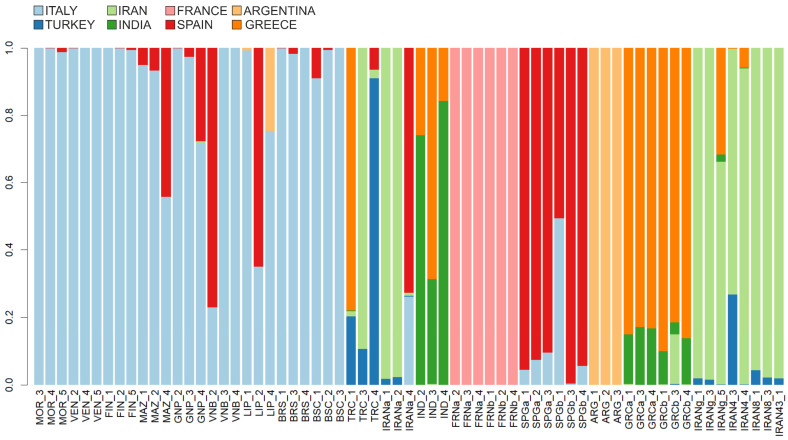
Membership probability plot of Discriminant Analysis of Principal Component performed retaining 34 PCs and 2 DFs.

**Figure 4 plants-14-00051-f004:**
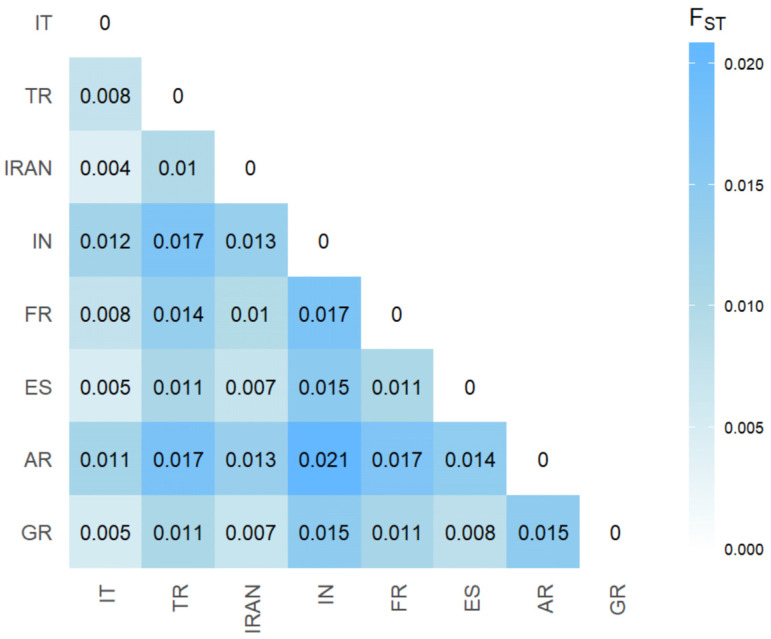
Matrix representing pairwise FST values between saffron of different geographical origin.

**Figure 5 plants-14-00051-f005:**
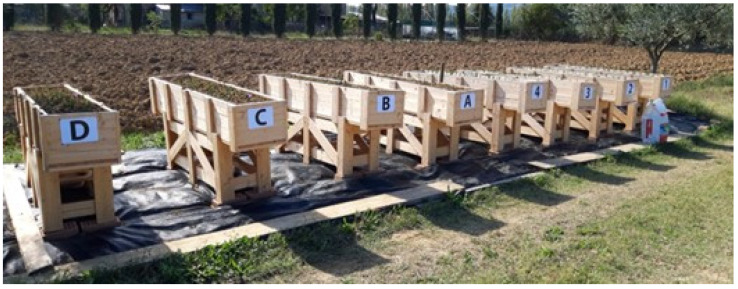
Wooden bins filled with organic soil (A–D) and farm soil (1–4).

**Table 1 plants-14-00051-t001:** (**a**) Combined analyses of variance of saffron traits of bulbs grown under different treatments in eight locations and for three years. (**b**) For these traits, data were collected only in seven locations.

(a)							
	Spice Yield (g/m^2^)			Flowers (No./m^2^)		Flowering (No. of Days)	
Source of Variation	Df	MS		MS		MS	
Rep (location year)	72	0.258	-	1075.8		5.5	
Treatments	2	6.163	*	7002.8	ns	126.9	*
Locations	7	2.750	ns	32,595.8	ns	1049.8	*
Years	2	17.337	***	105,236.0	***	444.4	***
Treatments × location	14	0.744	ns	10,740.3	**	43.3	ns
Treatments × year	4	0.866	***	3555.2	***	11.2	*
Location × year	14	2.146	***	22,505.0	***	335.0	***
Treatments × location × year	27	0.365	***	3511.0	***	26.1	***
Error	140	0.054		573.3		4.5	
Total	282						
**(b)**							
	**Corm Number** **(m^2^)**			**Corm Yield (g/m^2^)**		**Corm Weight (g)**	
**Source of Variation**	**Df**	**MS**		**MS**		**MS**	
Rep (location year)	63	340.5	-	145,089.0	-	3.88	-
Treatments	2	9838.3	ns	9,581,795.4	ns	694.82	**
Locations	6	40,258.5	ns	10,440,362.8	ns	534.37	ns
Years	2	22,248.5	***	16,512,813.3	***	381.23	***
Treatments × location	12	100,062.1	ns	3,831,779.7	ns	93.86	ns
Treatments × year	4	6292.4	***	1,663,513.8	***	13.49	*
Location × year	12	23,973.8	***	4,027,401.94	***	337.24	***
Treatments × location × year	20	5419.4	***	2,412,028.3	***	57.76	***
Error	112	380.6		162,896.0		4.20	
Total	233						

ns = not significant; *** = significant at *p* < 0.001; ** = significant at *p* < 0.01, * = significant at *p* < 0.05.

**Table 2 plants-14-00051-t002:** Mean value of saffron traits collected in different treatments, locations, and years.

	Spice Yield (g/m^2^) ^1^		Flowers (No./m^2^)		Flowering Length (days)		Bulbs (No./m^2^)		Bulbs (g/m^2^)	
**Treatments**										
Bin–farm soil	1.150	B	121.9		18.2	B	136.7		1919.2	
Bin–organic soil	1.377	A	138.5		19.4	B	129.4		2576.5	
Field (control)	0.849	C	126.2		22.2	A	121.1		2295.2	
**Locations**										
Alfonsi	0.898		102.6		31.7	A	90.0		2293.3	
Fontanelle	0.781		84.1		12.6	E	101.2		1930.1	
Mazzuoli	1.372		165.1		17.5	D	187.3		2116.1	
Porta Sole	1.373		161.5		22.1	B	63.6		1947.9	
Ro_Lo	0.927		106.2		18.2	CD	114.4		1878.1	
Venturi	1.546		164.7		19.1	C	131.0		2177.1	
Vinerbi	1.263		129.1		17.5	D	-		-	
Zafferano and Dintorni	0.849		110.0		20.2	C	118.3		1768.7	
**Years**										
2018	1.405	A	150.3	A	20.0	A	145.9	A	2882.8	A
2019	1.340	B	146.2	A	21.0	A	134.4	B	1926.0	B
2020	0.632	C	90.1	B	18.9	B	107.0	C	1982.1	B

^1^ For each source of variation and trait, means in a column followed by different letters are different for *p* < 0.05.

**Table 3 plants-14-00051-t003:** Combined analyses of variance of saffron quality spice (moisture content, bitterness, aromatic strength, and coloring strength) from trials based on different treatments in eight locations over three years.

	Moisture (%)			Bitterness			Aromatic Strength			Coloring Strength		
Source of Variation	Df	MS		Df	MS		Df	MS		Df	MS	
Rep (location)	15	1.32		15	118.8		15	14.1		15	880.5	
Treatments	2	0.94	*	2	30.2	ns	2	14.7	ns	2	203.4	ns
Locations	7	1.70	***	7	81.0	*	7	20.4	ns	7	327.3	ns
Treatments × location	14	0.34	ns	14	39.5	ns	14	16.9	ns	14	27.08	ns
Error	26	0.27		26	31.8		26	10.3		26	327.7	
Total	64			64			64			64		

ns = not significant; *** = significant at *p* < 0.001; * = significant at *p* < 0.05.

**Table 4 plants-14-00051-t004:** Mean value of saffron quality spice from trials based on different treatments in eight locations over three years.

	Moisture (%) ^1^		Bitterness		Aromatic Strength		Aromatic Strength	
Treatments								
Bin–organic soil (BO)	7.24	B	93.98		28.02		252.8	
Bin–farm soil (BF)	7.55	A	91.79		28.69		246.5	
Field (control)	7.56	A	91.94		29.75		248.5	
**Locations**								
Alfonsi	7.96	A	85.25	B	30.72	A	230.3	
Fontanelle	7.56	AB	99.20	A	28.67	E	264.5	
Mazzuoli	7.66	AB	90.78	B	28.00	D	252.2	
Porta Sole	7.43	AB	92.89	AB	28.11	B	248.3	
Ro_Lo	7.70	AB	97.00	A	28.72	CD	248.3	
Venturi	7.14	B	92.56	AB	29.67	C	247.7	
Vinerbi	6.58	C	92.56	AB	30.89	D	250.4	
Zafferano and Dintorni	7.58	AB	91.38	AB	25.78	C	252.2	

^1^ For each source of variation, means in a column followed by different letters are different for *p* < 0.05.

**Table 5 plants-14-00051-t005:** Geographic data of the farms that hosted the saffron trials with plants grown in wooden bins and the field from 2018 to 2020.

Farm Name	Location	Latitude	Longitude	Altitude (m asl)
Alfonsi	Nocera Umbra	43.1153	12.7698	520
Fontanelle	Marsciano	42.9070	12.3390	180
Mazzuoli	Città della Pieve	42.9546	12.0039	508
Porta Sole	Perugia	43.1280	12.4866	200
Ro_Lo	Foligno	42.9544	12.6995	234
Venturi	Gualdo Tadino	43.2288	12.7809	536
Vinerbi	Città della Pieve	42.9546	12.0039	508
Zafferano and Dintorni	Sant’Anatolia di Narco	42.7337	12.8355	328

## Data Availability

The data were deposited in the NCBI Sequence Read Archive (SRA) and are available at https://trace.ncbi.nlm.nih.gov/Traces/?view=study&acc=SRP546641, accessed on 18 November 2024 under the accession number SRP546641.

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
