# Peer review of "A Proposed Saffron Soilless Cultivation System for a Quality Spice as Certified by Genetic Traceability"

_plants, 2024, doi:10.3390/plants14010051_

Round 1
Reviewer 1 Report
Comments and Suggestions for Authors
This paper is based upon a well-designed, 3 year, research project with good controls and comparisons. It should be of strong interest to saffron producers, introducing a nice improvement in a growing application, showing that a peat enhanced medium, provides a production improvement over field-soil production.
Review comments and corrections:
Title: “ A Study Case in Central Italy” Does not seem necessary
L25: its’
L26: and..
L53: for its’ high quality
L101: Suggest: for bulbs …and for three years. Sig is not necessary as a table heading since significance is also indicated after each value , and explained as table sub heads.
Figure 1. A label should be provided for the vertical axis. I would also suggest a bar graph instead of a line graph, because there is no quantitative relationship among the treatments, each one is distinct.
Figure 2. What is the reason for including a smaller version of the scatter plot. The larger one is easy to read and show the relationships very nicely.
L176: Suggest: slightly more insight…
L232: threatening job… suggest: are laborious and not viewed as desirable by agricultural workers. due to the hard working conditions of the operators.
L238: placed in an open field…
L409: Briefly, …
Author Response
Comment 1: Title: “ A Study Case in Central Italy” Does not seem necessary
Response 1: We have modified the title accordingly.
Comment 2: L25: its’
Response 2: Done.
Comment 3: L26: and..
Response 3: Done.
Comment 4: L53: for its’ high quality
Response 4: Done.
Comment 5: L101: Suggest: for bulbs …and for three years. Sig is not necessary as a table heading since significance is also indicated after each value , and explained as table sub heads.
Response 5: Done in all Tables.
Comment 6: Figure 1. A label should be provided for the vertical axis. I would also suggest a bar graph instead of a line graph, because there is no quantitative relationship among the treatments, each one is distinct.
Response 6: Done.
Comment 7: Figure 2. What is the reason for including a smaller version of the scatter plot. The larger one is easy to read and show the relationships very nicely.
Response 7: The figure is the output of the software and the enlarged section is the magnification of one part.
Comment 8: L176: Suggest: slightly more insight…
Response 8: Done.
Comment 9: L232: threatening job… suggest: are laborious and not viewed as desirable by agricultural workers. due to the hard working conditions of the operators.
Response 9: Done.
Comment 10: L238: placed in an open field…
Response 10: Done.
Comment 11: L409: Briefly, …
Response 11: Done.
Reviewer 2 Report
Comments and Suggestions for Authors
In the manuscript named " A Proposed Saffron Soilless Cultivation System for a Quality Spice as Certified by Genetic Traceability: A Study Case in Central Italy", Mariani Alessandro et al have systematically evaluated saffron populations in Central Italy, their results would helpful for saffron production in Italy. However, there are some comments about this manuscript.
Major,
(1) The germplasms in present research were listed in table S1, but this table was missed, please check it.
(2) Key methods were missed in manuscript, including reference sequences in mapping process, SNP identified in detail, the population structure analysis, etc, please add these methods description in method section.
(3) The DNA-seq data or SNP genotype data (vcf file) would be released in public database or online site.
Minor,
(4) Discussion section, line 256 to line 274 could merged into single paragraph, not some points listed.
(5) Authors have evaluated some key traits, including number of flowers, in different regions, or different growth condition, and they have also identified these saffron population genetic information, could authors have tried to explore relationship between genetic and trait in saffron? In addition, how about suggestion about germplasm selection in Italy.
(6) The keyword, MCSeEd, was not suitable, please replace it with another word.
(7) The figure 3 was plotted based on DAPC analysis, not based on software STRUCTURE, or similar software, please check it. How about STRUCTURE analysis results, was it a similar results?
Author Response
Comment 1: The germplasms in present research were listed in table S1, but this table was missed, please check it.
Response 1: Done
Comment 2: Key methods were missed in manuscript, including reference sequences in mapping process, SNP identified in detail, the population structure analysis, etc, please add these methods description in method section.
Response 2: In the Materials and methods, the paragraph 4.2.2. has been rewritten with more details on mapping process and SNP calling along with functions of adegenet and poppr R packages employed in the population genetic analysis.
Comment 3: The DNA-seq data or SNP genotype data (vcf file) would be released in public database or online site.
Response 3: The DNA-seq data were already submitted but the NCBI link did not work properly. We have now replaced it with the direct link as follow:
Data Availability Statement: The data have been deposited in the NCBI Sequence Read Archive (SRA) and are available at https://trace.ncbi.nlm.nih.gov/Traces/?view=study&acc=SRP546641 under the accession number SRP546641.
Comment 4: Discussion section, line 256 to line 274 could merged into single paragraph, not some points listed.
Response 4: Done
Comment 5: Authors have evaluated some key traits, including number of flowers, in different regions, or different growth condition, and they have also identified these saffron population genetic information, could authors have tried to explore relationship between genetic and trait in saffron?
Response 5: The corms used in the open field trials were all certified and provided by Mazzuoli Farm (see end of Paragraph 1 of 4.4.1). those used for the genetic characterization were provided by each farm, with the objective to assess their traceability.
Comment 5 bis: In addition, how about suggestion about germplasm selection in Italy.
Response 5bis: The objective of this research was not selection, rather trying to suggest new techniques.
Comment 6: The keyword, MCSeEd, was not suitable, please replace it with another word.
Response 6: MCSeEd has been removed and replaced with Local Varieties
Comment 7: The figure 3 was plotted based on DAPC analysis, not based on software STRUCTURE, or similar software, please check it. How about STRUCTURE analysis results, was it a similar results?
Response 7: For our genetic population analysis, we chose to use the adegenet R package and performed a DAPC test to infer the structure of the saffron population. This decision was primarily driven by the output provided by DAPC, which includes a discriminant space where genetic groups are visualized as clusters, membership probabilities for individuals within each cluster, and results that can be displayed using scatterplots or compoplots. Additionally, DAPC is computationally efficient, making it particularly suitable for analyzing large datasets and exploring population structure without the need for complex simulations.
Round 2
Reviewer 2 Report
Comments and Suggestions for Authors
Thanks for authors’ works, the review had been well revised, most of my comments were well addressed in revision. However, some misspelt were still present in manuscript, for example, there were a “-” after figures, the conclusion section title was also a misspelt, etc., please carefully check them. I have no new comments about it, good luck.
Author Response
Comment 1: some misspelt were still present in manuscript, for example, there were a “-” after figures, the conclusion section title was also a misspelt, etc., please carefully check them.
Answer 1: the suggested corrections were made and also the manuscript was carefully revised